# Gender Differences in Complete Blood Count and Inflammatory Ratios among Patients with Bipolar Disorder

**DOI:** 10.3390/brainsci11030363

**Published:** 2021-03-12

**Authors:** Laura Fusar-Poli, Andrea Amerio, Patriciu Cimpoesu, Pietro Grimaldi Filioli, Antimo Natale, Guendalina Zappa, Eugenio Aguglia, Mario Amore, Gianluca Serafini, Andrea Aguglia

**Affiliations:** 1Psychiatry Unit, Department of Clinical and Experimental Medicine, University of Catania, 95123 Catania, Italy; laura.fusarpoli@gmail.com (L.F.-P.); antimo.natale88@gmail.com (A.N.); eugenio.aguglia@unict.it (E.A.); 2Section of Psychiatry, Department of Neuroscience, Rehabilitation, Ophthalmology, Genetics, Maternal and Child Health, University of Genoa, 16126 Genoa, Italy; andrea.amerio@unige.it (A.A.); patriciu.cimpoesu@gmail.com (P.C.); piotr.grimaldi@gmail.com (P.G.F.); guendalina.zap@gmail.com (G.Z.); mario.amore@unige.it (M.A.); gianluca.serafini@unige.it (G.S.); 3IRCCS Ospedale Policlinico San Martino, 16132 Genoa, Italy; 4Department of Psychiatry, Tufts University, Boston, MA 02110, USA

**Keywords:** gender, bipolar disorder, inflammation, biomarkers, red blood cells, platelets, monocyte, eosinophils, neutrophil-to-lymphocyte, monocyte-to-lymphocyte, platelet-to-lymphocyte

## Abstract

**Background:** Evidence suggested that inflammation may be involved in the etiopathogenesis of bipolar disorder (BD), a chronic psychiatric condition affecting around 2–3% of the general population. However, little is known regarding potential gender differences in peripheral biomarkers of BD, such as neutrophil-to-lymphocyte (NLR), platelet-to-lymphocyte (PLR), and monocyte-to-lymphocyte (MLR) ratios. **Methods:** In total, 197 females and 174 males with BD in different phases (i.e., (hypo)mania, depression, euthymia) were recruited. A blood sample was drawn to perform a complete blood count (CBC). NLR, PLR, and MLR were subsequently calculated, and differences were computed according to the illness phase and gender. **Results:** PLR was consistently higher in (hypo)manic than depressed patients, in both males and females. No significant gender differences in PLR value were found when considering only (hypo)mania. Conversely, NLR was increased in (hypo)mania only among males, and gender differences were retrieved in the (hypo)manic subgroup. The findings related to MLR were only marginally significant. Higher platelets values were associated with (hypo)mania only in the female group. Basophils and eosinophils appeared gender- but not state-dependent. **Conclusions:** Our findings provide further evidence that increased PLR levels may be associated with (hypo)mania in bipolar patients, regardless of gender. Moreover, the usefulness of NLR as a peripheral biomarker of BD appeared limited to males while the role of platelets to females. As CBC represents a low-cost and easily accessible test, researchers should investigate in-depth its potential usefulness as a biomarker of BD and other psychiatric disorders.

## 1. Introduction

Bipolar disorder (BD) is a severe affective disease characterized by the presence of recurring manic or hypomanic phases that may alternate with major depressive episodes (MDE) [1]. BD represents a leading cause of burden worldwide and contributes significantly to premature death [2]. Subjects with BD are at high risk of presenting with co-occurrent psychiatric illnesses, including alcohol and substance use disorders [3,4,5], as well as serious medical comorbidities [3,6,7,8,9,10]. Moreover, suicide risk is up to 30 times higher in individuals with BD than the general population, with one in four to five subjects attempting suicide lifetime [11,12]. 

Epidemiological studies estimated the prevalence of bipolar and related disorders to be around 2–3% in the general population, with no significant gender differences [13,14]. Nevertheless, several reports suggested that the symptomatic presentation of bipolar patients may show differences between males and females. For instance, females seem to have a higher incidence of BD type II diagnosis and hypomania than males [13,15,16,17,18], and present more frequently with seasonal and rapid cycling patterns [19,20,21,22] and mixed episodes [15,23,24,25,26,27]. Moreover, they seem more likely to suffer from subclinical depressed mood and dysphoria [15,22,28,29], although the number of MDE and the time spent in major depression would not differ across gender [15,16,24,25,30]. 

Even if the etiopathogenesis of BD has not been clarified yet, an interaction between genetic and environmental risk factors seems highly plausible. The literature extensively demonstrated that the immune and inflammatory systems may play a crucial role in determining the onset and recurrence of major depressive disorder (MDD) [31,32]. Thus, the hypothesis of neuroinflammatory mechanisms in the etiology of BD has gained growing credibility [33,34]. For instance, several peripheral inflammatory biomarkers (i.e., cytokines) appear altered in patients with BD [35,36]. Among these, a growing interest has been acquired by inflammatory ratios, i.e., neutrophil-to-lymphocyte (NLR), monocyte-to-lymphocyte (MLR), and platelet-to-lymphocyte (PLR) ratios. These variables are derived from complete blood count (CBC), a low-cost and simple examination widespread in routine clinical practice [37]. According to recent studies, inflammatory ratios are significantly increased in BD [38], with higher values in (hypo)manic than MDE [39,40,41], thus representing potential state markers of this condition.

Indeed, the gender differences observed in the clinical presentation of mood disorders may be partially explained by the alterations of the inflammatory and immune system [35,42]. Despite the growing interest in the topic, only few researchers have investigated the potential gender differences in inflammatory biomarkers in bipolar patients. According to a recent review [35], c-reactive protein (CRP) [43,44], interleukin (IL)-6 and IL-18 [45], soluble IL-2 receptor (sIL-2R), secretory tumor necrosis factor-alpha (sTNF-α) [46], and TNF-α [45,46] would not differ between males and females with BD. Conversely, significant gender differences were found in white blood cells (WBC), as deviations in this count appeared associated with a greater severity of specific depressive symptoms, with more robust findings among males [47]. Other researchers found a positive correlation between the levels of highly sensitive CRP (hsCRP) and number of manic episodes and MDE among females with BD. The levels of hsCRP and IL-6 were also positively correlated with current manic symptoms. The findings were not replicated in bipolar males [48], suggesting that the inflammatory framework might differ across gender. To the best of our knowledge, no studies have specifically examined gender differences in inflammatory ratios among bipolar patients.

## 2. Material and Methods

### 2.1. Participants

We conducted a cross-sectional study including in- and outpatients with a primary diagnosis of BD. Participants were consecutively referred to the Section of Psychiatry, Department of Neuroscience, Rehabilitation, Ophthalmology, Genetics, Maternal and Child Health (DINOGMI), IRCCS Ospedale Policlinico San Martino, University of Genoa (Italy), and Psychiatry Unit, Department of Clinical and Experimental Medicine, Gaspare Rodolico Hospital, University of Catania (Italy), in the period between January 2017 and June 2020. 

To be included into the present study, patients had to fulfil the following inclusion criteria: (a) being hospitalized in a psychiatric unit with a primary diagnosis of BD for a current (hypo)manic or MDE, or being followed by an outpatient service in the case of euthymic patients; (b) being 18 years or older; (c) providing written informed consent to participate in the study. Clinically, euthymia has been defined using specific psychometric criteria of < 10 on the Montgomery–Åsberg Depression Rating Scale (MADRS) [49] and < 7 on the Young Mania Rating Scale (YMRS) [50].

The exclusion criteria were as follows: (a) pregnancy or recent childbirth, (b) severe and/or acute medical comorbidities or any other conditions that could affect the measured parameters, (c) a current substance use disorder, including alcohol, (d) having a history of acute neurological injury or a diagnosis of intellectual disability, and (e) the inability or refusal to provide a written informed consent to participate in the study. 

Potential participants were given a full explanation of the study’s aims and procedures and opportunity to ask questions. Written informed consent was obtained from all participants according to the current version of the Declaration of Helsinki. The study design was approved by the “IRCCS Ospedale Policlinico San Martino” Ethical Review Board (82/13, amended on 10 February 2017, and registered with number 028 of 2 March 2017).

### 2.2. Assessments and Procedures

Basic socio-demographic and clinical features were obtained through the administration of a semi-structured interview including age, gender, marital and occupational status, education level, psychiatric diagnosis, and pharmacological treatment. Psychiatric diagnoses were based on the criteria of the Diagnostic and Statistical Manual of Mental Disorders, fifth edition (DSM-5), and were formulated by expert psychiatrists within the in- and outpatient clinical settings [1].

After a 12 h overnight fast, 3 mL of blood were collected from a forearm vein of participants in hemogram tubes containing EDTA. Samples were processed within 30 min after collection with flow cytometry in the respective laboratory sections (IRCCS Ospedale Policlinico San Martino, Genoa, and Policlinico “G. Rodolico”, Catania). The following counts were evaluated: red blood cells (RBC), hemoglobin, hematocrit, mean corpuscular volume (MCV), mean corpuscular hemoglobin (MCH), mean corpuscular hemoglobin concentration (MCHC), red blood cell distribution width coefficient of variation (RDW CV), neutrophils, lymphocytes, monocytes, eosinophils, basophils, and platelets. PLR, NLR and MLR were considered as peripheral inflammatory biomarkers and calculated. 

### 2.3. Statistical Analysis

Continuous and categorical variables were presented as means and standard deviations (SD) or frequency and percentage, respectively. Normal distribution was assessed using the Kolmogorov–Smirnov test, before applying statistical analyses.

First, the sample was divided according to gender, i.e., males and females. One-way ANOVAs were then performed to investigate differences in CBC and inflammatory ratios according to the different mood episodes in each of the two groups, separately. Second, the sample was divided into three subgroups according to bipolar illness phase, i.e., (hypo)manic episode, MDE, and euthymia. Student’s *t*-test for independent samples was performed to evaluate gender differences in each group.

Statistical analyses were performed using the Statistical Package for Social Sciences (SPSS) for Windows 25.0 (IBM Corp., Armonk, NY, USA). Results were regarded as statistically significant for *p* < 0.05.

## 3. Results

### 3.1. Characteristics of the Sample

The sample included in the present study comprised 371 patients with BD, of which 53.1% (*n* = 197) were females. The mean age of participants was 51.91 ± 13.88 years and on average they received 11.97 years of education. The subjects were mainly single (47.2%) and unemployed (65%). Among bipolar patients, 143 were experiencing a (hypo)manic episode, 151 a MDE, and 77 were in a euthymic phase. More details about socio-demographic characteristics and medications taken by the participants are reported in Table 1.

### 3.2. Differences in Complete Blood Count and Inflammatory Ratios According to the Phase of Bipolar Illness in Males and Females, Separately

First, we divided the sample according to gender. Considering the male subsample, significant differences were found in two of the three inflammatory ratios of interest. Specifically, the NLR and PLR were higher in (hypo)manic than depressed men with BD (*p* = 0.005 and *p* = 0.006, respectively). No other statistically significant differences were retrieved in the CBC of the male group. CBC parameters and inflammatory ratios in males with BD are presented in Table 2.

Examining the sample of females with BD, some differences in white blood cells (WBC), specifically eosinophils and basophils, have been observed. In addition, platelets appeared significantly elevated in (hypo)mania, compared to both depression (*p* = 0.004) and euthymia (*p* = 0.017). As per RBC, only MCHC was significantly reduced in (hypo)mania compared to euthymia. Considering inflammatory ratios, instead, a difference was found only in PLR, which appeared significantly higher in (hypo)mania than both MDE (*p* = 0.002) and euthymia (*p* = 0.009). CBC parameters and inflammatory ratios in females with BD are presented in Table 3.

### 3.3. Gender Differences in Cell Blood Count and Inflammatory Ratios According to the Bipolar Illness Phase

As a second step, we divided the sample according to the different illness phases (i.e., (hypo)mania, MDE, and euthymia), and analyzed gender differences in each of the three subgroups. A statistically significant difference was found in the monocytes in all groups, with males having higher mean values than females. In addition, eosinophils were increased in males in the MDE and euthymic groups, and neutrophils were higher in euthymic males than females. A statistically significant difference in RBC and related parameters was found in each of the three subgroups. As per the inflammatory ratios, a statistically significant difference was found only in the group of patients experiencing a (hypo)manic episode, with males having significantly higher NLR and MLR values than females. The results of Student’s *t*-tests are reported in Table 4.

The means and standard deviations of the three inflammatory ratios (NLR, PLR, and MLR) are presented in Figure 1, divided by gender and bipolar illness phase.

## 4. Discussion

BD is a complex psychiatric condition with multiple etiological factors interacting among each other. A large amount of evidence suggested that neuroinflammation is implicated in the etiopathology and course of BD [51,52]. Thus, peripheral inflammatory parameters have been proposed as potential biomarkers of BD and its different phases. However, only few studies examined gender as a factor, rather than controlling for it as a covariate when exploring biomarkers [35]. To our knowledge, this is the first study to specifically investigate gender differences in CBC as well as inflammatory ratios (i.e., NLR, PLR, and MLR) in patients with BD across different illness phases.

Our findings primarily confirm that several inflammatory ratios are consistently more elevated in (hypo)mania than MDE, further supporting the results of recent cross-sectional studies [39,40]. First, PLR appeared significantly increased in (hypo)mania without any differences between males and females. This is in line with previous reports [40,41], in which PLR was uniquely associated with (hypo)mania. Since no gender difference was observed in PLR values while considering only the (hypo)manic subgroup of bipolar patients, it is possible to hypothesize that PLR may represent the strongest biomarker of (hypo)mania among the considered inflammatory ratios, regardless of gender. However, it is worth mentioning that this hypothesis contrasts a recent cross-sectional investigation in which no differences in PLR were found across different mood episodes [39].

Conversely, NLR was significantly increased only in (hypo)manic males, while the female sample did not show any difference across the different bipolar phases. In fact, after dividing the sample according to the bipolar illness phases, a significant gender difference was found in the group of (hypo)manic patients, with females having significantly lower NLR means than males. Thus, we might assume that both gender and mood episode may impact the NLR value, which may be considered a gender-dependent peripheral biomarker of (hypo)mania in BD.

As for MLR, we only found a marginally significant gender difference in the (hypo)manic group (*p* = 0.045). Moreover, while considering males and females separately, no differences were found across the different mood states. To our knowledge, this ratio has been explored by few studies so far, with contrasting results [39,40,41,53,54]. Therefore, its significance as a peripheral biomarker in BD remains uncertain, and further studies are needed to elucidate its role.

Conversely, the role of monocytes per se attracts some interest; in fact, they were consistently increased in males but did not differ across different bipolar illness phases. Similar results were retrieved for eosinophils, although gender difference was limited to the depressive and euthymic phases. Interestingly, platelets were significantly increased in (hypo)manic females compared to those experiencing a MDE or euthymia. This finding was not replicated in the male sample. However, after dividing the whole sample into different phases, gender differences in platelet concentration were retrieved only in the euthymic state.

Finally, gender differences were found also in other CBC parameters. For instance, RBC, hematocrit, and hemoglobin were consistently lower in females than males, regardless of bipolar illness phase. However, these findings are likely related to physiological gender differences, and do not appear clinically relevant [55,56]. 

This is the first study to investigate gender differences in CBC as well as NLR, PLR, and MLR in patients with BD, according to different illness phases. Nevertheless, several limitations should be acknowledged while discussing our results. First, although we compared patients experiencing different phases of BD, we did not include a sample of healthy participants in our study. Second, we did not conduct a power analysis; therefore, the sample size might have been not sufficiently large to detect significant differences. Particularly, the group of participants in the euthymic phase was the smallest. Third, concomitant physiological (e.g., menstrual cycle, eating habits) or pathological (e.g., obesity, cancer, autoimmune disorders) conditions potentially influencing the inflammatory status of participants were not considered as confounding factors in our analyses, nor did we adjust for age. Fourth, some important peripheral biomarkers of neuroinflammation (e.g., interleukins, chemokines, TNF-α) were not taken into account, as they are more expensive and infrequently used in the clinical practice. In addition, blood parameters were not correlated with the severity of symptoms, appropriately measured with validated scales. Finally, the cross-sectional nature of the present study did not allow us to follow up individuals across different phases of bipolar illness.

In spite of the aforementioned limitations, the present study may find potential applications in clinical practice and provide numerous cues for future research. CBC is a simple and low-cost measure routinely performed in psychiatric practice, both in in- and outpatient services [37]. In fact, CBC values are regularly monitored by clinicians to evaluate the potential side effects of drugs and general health status of bipolar patients. Given the growing body of research linking neuroinflammation to specific illness phases in BD, CBC as well as inflammatory ratios (particularly PLR or NLR) could be adopted by clinicians to monitor the clinical course of patients with BD. It is also likely that neuroinflammatory parameters may be associated with symptoms severity: future cross-sectional and longitudinal studies should be conducted to correlate inflammatory overexpression with clinical symptoms in BD through the use of validated psychometric instruments. Other variables such as CRP or the erythrocyte sedimentation rate might be combined. Moreover, investigating the interaction between inflammatory parameters and hormonal levels may further help to clarify gender differences in BD.

## 5. Conclusions

In conclusion, our study provides further evidence regarding the utility of inflammatory ratios and other CBC parameters in predicting mood episodes in BD. Moreover, it expands on previous knowledge about biological gender differences. Future research may help clinicians and researchers to identify some biological phenotypes of this complex condition.

## Figures and Tables

**Figure 1 brainsci-11-00363-f001:**
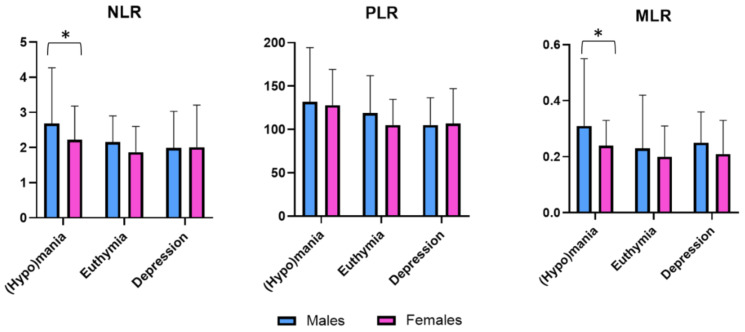
Gender differences in neutrophil-to-lymphocyte (NLR), platelet-to-lymphocyte (PLR), and monocyte-to-lymphocyte (MLR) ratios according to bipolar state. * Statistically significant difference (*p* < 0.05).

**Table 1 brainsci-11-00363-t001:** Characteristics of the total sample (*n* = 371).

Gender (female), *n* (%)	197 (53.1)
Age (years), mean ± SD	51.91 ± 13.88
Years of education, mean ± SD	11.97 ± 2.18
Marital status, *n (%)*	
Single	175 (47.2)
Married	101 (27.2)
Divorced	65 (17.5)
Widowed	30 (8.1)
Employed, *n (%)*	130 (35.0)
Illness phase, *n (%)*	
(Hypo)manic episode	143 (38.5)
Major depressive episode	151 (40.7)
Euthymic phase	77 (20.8)
Pharmacological treatment (*n* = 313)	
Antidepressants, *n* (%)	137 (43.8)
Mood stabilizers, *n* (%)	
Valproate	134 (42.8)
Lithium	132 (42.2)
Others	77 (24.6)
Antipsychotics, *n* (%)	268 (85.6)
Typical	53 (16.9)
Atypical	246 (78.6)
Long-acting injection	13 (4.2)
Benzodiazepines, *n* (%)	231 (73.8)
Number of medications, mean ± SD	3.90 ± 1.19

**Table 2 brainsci-11-00363-t002:** Differences in complete blood count and inflammatory ratios in bipolar males, according to illness phase.

Mean ± SD	Male Gender			*F*	*p*
	(Hypo)manic Episode (*n* = 73)	Major Depressive Episode (*n* = 64)	Euthymic Phase (*n* = 37)		
Neutrophils	5.12 ± 1.84	4.44 ± 1.70	4.77 ± 1.44	2.693	0.071
Lymphocytes	2.22 ± 0.85	2.39 ± 0.71	2.33 ± 0.67	0.867	0.422
Monocytes	0.65 ± 0.44	0.57 ± 0.21	0.68 ± 0.26	1.299	0.276
Eosinophils	0.20 ± 0.15	0.23 ± 0.13	0.23 ± 0.16	0.980	0.378
Basophils	0.04 ± 0.02	0.04 ± 0.03	0.05 ± 0.04	1.146	0.321
Platelets	259.25 ± 65.94	239.45 ± 67.24	258.19 ± 57.57	1.826	0.164
NLR	2.68 ± 1.59	1.99 ± 1.04	2.16 ± 0.74	5.506	0.005 *
PLR	132.11 ± 62.24	105.07 ± 31.54	119.16 ± 42.95	5.224	0.006 *
MLR	0.31 ± 0.24	0.25 ± 0.11	0.23 ± 0.19	2.829	0.062
Red blood cell	4.91 ± 0.65	4.77 ± 0.55	4.90 ± 0.43	1.136	0.323
Hemoglobin	142.99 ± 17.46	142.94 ± 15.32	145.27 ± 13.34	0.307	0.736
Hematocrit	42.66 ± 4.54	42.91 ± 4.40	43.33 ± 4.19	0.277	0.758
MCV	87.72 ± 7.81	90.31 ± 5.51	88.55 ± 6.59	2.522	0.083
MCH	29.33 ± 3.01	30.08 ± 2.06	29.71 ± 2.35	1.457	0.236
MCHC	334.07 ± 11.32	333.02 ± 10.09	335.57 ± 8.47	0.719	0.489
RDW CV	13.83 ± 1.30	13.62 ± 1.02	13.42 ± 1.43	1.410	0.247

* Bonferroni post-hoc: NLR and PLR significantly different in (hypo)manic vs. major depressive episode (*p* < 0.05). MCH: mean corpuscolar hemoglobin; MCHC: mean corpuscolar hemoglobin concentration; MCV: mean corpuscolar volume; MLR: monocyte-to-lymphocyte ratio; NLR: neutrophil-to-lymphocyte ratio; PLR: platelet-to-lymphocyte ratio; RDW CV: red blood cell distribution width coefficient of variation.

**Table 3 brainsci-11-00363-t003:** Differences in complete blood count and inflammatory ratios in bipolar females, according to illness phase.

Mean ± SD	Female Gender			*F*	*p*
	(Hypo)manic Episode (*n* = 70)	Major Depressive Episode (*n* = 87)	Euthymic Phase (*n* = 40)		
Neutrophils	4.62 ± 1.92	4.33 ± 1.45	4.04 ± 1.34	1.671	0.191
Lymphocytes	2.19 ± 0.70	2.35 ± 0.69	2.34 ± 0.92	0.983	0.376
Monocytes	0.52 ± 0.15	0.49 ± 0.20	0.53 ± 0.19	0.626	0.536
Eosinophils	0.22 ± 0.14	0.19 ± 0.13	0.15 ± 0.07	3.455	0.034 *
Basophils	0.03 ± 0.02	0.04 ± 0.03	0.03 ± 0.03	3.236	0.042 *
Platelets	263.33 ± 72.08	231.25 ± 52.20	229.55 ± 57.44	6.463	0.002 *
NLR	2.22 ± 0.96	2.01 ± 1.20	1.87 ± 0.73	1.626	0.199
PLR	128.07 ± 41.16	106.77 ± 40.38	105.07 ± 29.71	7.181	0.001 *
MLR	0.24 ± 0.09	0.21 ± 0.12	0.20 ± 0.11	2.706	0.069
Red blood cell	4.44 ± 0.43	4.44 ± 0.49	4.53 ± 0.52	0.495	0.610
Hemoglobin	128.57 ± 14.22	128.47 ± 13.00	133.25 ± 13.59	1.943	0.146
Hematocrit	39.46 ± 3.95	39.11 ± 3.54	39.98 ± 3.71	0.756	0.471
MCV	88.98 ± 6.03	88.69 ± 8.83	88.92 ± 7.10	0.033	0.968
MCH	29.00 ± 2.41	29.12 ± 3.28	29.61 ± 2.64	0.601	0.549
MCHC	325.73 ± 11.56	328.10 ± 10.33	333.00 ± 10.24	5.826	0.003 *
RDW CV	13.86 ± 1.34	14.32 ± 1.87	13.67 ± 0.95	3.054	0.049 *

* Bonferroni post-hoc: Platelets and PLR significantly different in (hypo)manic vs. major depressive episode (*p* < 0.05); Eosinophils, platelets, PLR and MCHC significantly different in (hypo)manic episode vs. euthymia (*p* < 0.05). MCH: mean corpuscolar hemoglobin; MCHC: mean corpuscolar hemoglobin concentration; MCV: mean corpuscolar volume; MLR: monocyte-to-lymphocyte ratio; NLR: neutrophil-to-lymphocyte ratio; PLR: platelet-to-lymphocyte ratio; RDW CV: red blood cell distribution width coefficient of variation.

**Table 4 brainsci-11-00363-t004:** Gender differences in cell blood count values during different bipolar illness phase.

	(Hypo)manic Episode		Major Depressive Episode		Euthymic Phase	
	*t°*	*p*	*t*°	*p*	*t*°	*p*
Neutrophils	1.589	0.114	0.418	0.676	2.304	0.024 *
Lymphocytes	0.235	0.814	0.366	0.715	–0.040	0.968
Monocytes	2.235	0.027 *	2.236	0.027 *	2.498	0.015 *
Eosinophils	–0.504	0.615	2.081	0.039 *	2.834	0.006 *
Basophils	1.380	0.170	–0.794	0.429	1.775	0.081
Platelets	–0.353	0.724	0.844	0.400	2.183	0.032 *
NLR	2.061	0.041 *	–0.132	0.895	1.728	0.088
PLR	0.455	0.649	–0.281	0.779	1.685	0.096
MLR	2.027	0.045 *	1.701	0.091	0.842	0.402
Red blood cells	5.065	<0.001 *	3.892	<0.001 *	3.416	0.001 *
Hemoglobin	5.401	<0.001 *	6.263	<0.001 *	3.911	<0.001 *
Hematocrit	4.488	<0.001 *	5.884	<0.001 *	3.720	<0.001 *
MCV	–1.076	0.284	1.297	0.197	–0.237	0.813
MCH	0.725	0.469	2.051	0.042 *	0.172	0.864
MCHC	4.358	<0.001 *	2.936	0.004 *	1.194	0.236
RDW CV	–0.154	0.878	–2.695	0.008 *	–0.917	0.362

* Statistically significant difference (*p* < 0.05). ° *t* > 0 are associated with higher mean values in males. NLR: neutrophil-to-lymphocyte ratio; PLR: platelet-to-lymphocyte ratio; MLR: monocyte-to-lymphocyte ratio; MCV: mean corpuscolar volume; MCH: mean corpuscolar hemoglobin; MCHC: mean corpuscolar hemoglobin concentration; RDW CV: red blood cell distribution width coefficient of variation.

## Data Availability

The data presented in this study are available on request from the corresponding author. The data are not publicly available due to privacy/ethical restrictions.

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
