# Peer review of "Gender Differences in Complete Blood Count and Inflammatory Ratios among Patients with Bipolar Disorder"

_brainsci, 2021, doi:10.3390/brainsci11030363_

Round 1

Reviewer 1 Report

The topic of this manuscript is very interesting for the field and in this original paper, the authors conducted a study to assess gender differences in neutrophil-to-lymphocyte (NLR), platelet-to-lymphocyte (PLR), and monocyte-to-lymphocyte (MLR) ratios in patients with bipolar disorder. Please find below a few suggestions: 1) The study did not include a control group to compare with the bipolar group, which would strengthen the findings. This is a limitation in the study, so please considering discuss this in the limitation section. 2) In Figure 1 it is not shown the statistical significance in the graphs. Based on the text, it should show that ''a statistically significant difference was found only in the group of patients experiencing a (hypo)manic episode, with males having significantly higher NLR and MLR values than females'' (line 193-195). 3) In the caption of Figure 1, please also add the meaning of the abbreviations. 4) In the discussion section be careful with sentences like ‘Our findings strengthen the hypothesis that PLR may represent a predictor of (hypo)mania (line 253)’. Considering the nature of the study and the analysis conducted this sentence is speculative and I suggest rewriting it.

Author Response

Reviewer 1

The topic of this manuscript is very interesting for the field and in this original paper, the authors conducted a study to assess gender differences in neutrophil-to-lymphocyte (NLR), platelet-to-lymphocyte (PLR), and monocyte-to-lymphocyte (MLR) ratios in patients with bipolar disorder. Please find below a few suggestions

  • RESPONSE: We thank the Reviewer for the efforts made for revising our manuscript and for the insightful suggestions. We have tried to do our best to address the concerns raised by the Reviewer. We hope that our manuscript has improved after addressing their issues. Please, find below our replies.

 Question 1: The study did not include a control group to compare with the bipolar group, which would strengthen the findings. This is a limitation in the study, so please considering discuss this in the limitation section.

  • R1: We thank the Reviewer for the comment. We agree that this may represent a potential limitation and therefore we have added a sentence in dedicated paragraph:“First, although we compared patients experiencing different phases of BD, we did not include a sample of healthy participants in our analyses.”

Question 2: In Figure 1 it is not shown the statistical significance in the graphs. Based on the text, it should show that ''a statistically significant difference was found only in the group of patients experiencing a (hypo)manic episode, with males having significantly higher NLR and MLR values than females'' (line 193-195).

  • R2: We thank the Reviewer for noticing this shortcoming. We have addressed the issue and hope that Figure 1 is now clearer for the readers.

Question 3: In the caption of Figure 1, please also add the meaning of the abbreviations.

  • R3: Added following the Reviewer suggestion.

Question 4: In the discussion section be careful with sentences like ‘Our findings strengthen the hypothesis that PLR may represent a predictor of (hypo)mania (line 253)’. Considering the nature of the study and the analysis conducted this sentence is speculative and I suggest rewriting it.

  • R4: We thank the Reviewer for the suggestion. We agree that the sentence might be too speculative and therefore we have deleted it.

ADDITIONAL CHANGES

Due to the cross-sectional nature of our study, we have deleted throughout the text the concept of “predictor” and introduced instead the concept of “association” between mood state/gender and specific variables.

Reviewer 2 Report

Interesting work dealing with important elements of inflammation in patients with mental disorders. However, it requires a few adjustments. First of all, it should be noted in the introduction that in unipolar depression, the inflammatory mechanism is also clearly described and may be associated with similar mechanisms (e.g. in the works: Galecka et.all.doi: 10.3390 / jpm11020066 and 10.1007 / s43440-020-00202-2) ). I think 2-3 sentences will be enough. Additionally, the study groups are not very numerous and the statistical analysis is not very strong. Therefore, at the end of the discussion, the related limitations should be noted. 

Author Response

Reviewer 2

Interesting work dealing with important elements of inflammation in patients with mental disorders. However, it requires a few adjustments.

  • RESPONSE: We thank the Reviewer for the efforts made for revising our manuscript and for the insightful suggestions. We have tried to do our best to address the concerns raised by the Reviewer. We hope that our manuscript has improved after addressing their issues. Please, find below our replies.

Question 1: First of all, it should be noted in the introduction that in unipolar depression, the inflammatory mechanism is also clearly described and may be associated with similar mechanisms (e.g. in the works: Galecka et al. doi: 10.3390 / jpm11020066 and 10.1007 / s43440-020-00202-2) ). I think 2-3 sentences will be enough. 

  • R1: We thank the Reviewer for the comment and for suggesting the references that may help to strengthen our hypotheses. As suggested, we have added a few sentences in the Introduction, introducing the references suggested by the Reviewer.

“Even if the etiopathogenesis of BD has not been clarified yet, an interaction be-tween genetic and environmental risk factors seems highly plausible. Literature has extensively demonstrated that the immune and inflammatory system may play a crucial role in determining the onset and recurrence of major depressive disorder (MDD) [31,32]. Thus, the hypothesis on the implication of neuroinflammation in the etiology of BD has gained growing credibility [33,34].“

Question 2: Additionally, the study groups are not very numerous and the statistical analysis is not very strong. Therefore, at the end of the discussion, the related limitations should be noted. 

  • R2: Thank you for the comment. We have added this limitation in the dedicated section, as reported below:

“Second, as we did not conduct a power analysis, the sample size might have been not sufficiently large to detect significant differences. Particularly, the group of participants in the euthymic phase was the smallest.”

ADDITIONAL CHANGES

Due to the cross-sectional nature of our study, we have deleted throughout the text the concept of “predictor” and introduced instead the concept of “association” between mood state/gender and specific variables.

Round 2

Reviewer 2 Report

The authors have introduced comments. A newsworthy article.